# VARIATIONAL WASSERSTEIN GRADIENT FLOW

## ABSTRACT

The gradient flow of a function over the space of probability densities with respect to the Wasserstein metric often exhibits nice properties and has been utilized in several machine learning applications. The standard approach to compute the Wasserstein gradient flow is the finite difference which discretizes the underlying space over a grid, and is not scalable. In this work, we propose a scalable proximal gradient type algorithm for Wasserstein gradient flow. The key of our method is a variational formulation of the objective function, which makes it possible to realize the JKO proximal map through a primal-dual optimization. This primal-dual problem can be efficiently solved by alternatively updating the parameters in the inner and outer loops. Our framework covers all the classical Wasserstein gradient flows including the heat equation and the porous medium equation. We demonstrate the performance and scalability of our algorithm with several numerical examples.

## 1 INTRODUCTION

The Wasserstein gradient flow models the gradient dynamics over the space of probability densities with respect to the Wasserstein metric. It was first discovered by Jordan, Kinderlehrer, and Otto (JKO) in their seminal work (Jordan et al., 1998). They pointed out that the Fokker-Planck equation is in fact the Wasserstein gradient flow of the free energy, bringing tremendous physical insights to this type of partial differential equations (PDEs). Since then, the Wasserstein gradient flow has played an important role in optimal transport, PDEs, physics, machine learning, and many other areas (Ambrosio et al., 2008; Otto, 2001; Adams et al., 2011; Santambrogio, 2017; Carlier et al., 2017; Frogner & Poggio, 2020).

Despite the abundant theoretical results on the Wasserstein gradient flow established over the past decades (Ambrosio et al., 2008; Santambrogio, 2017), the computation of it remains a challenge. Most existing methods are either based on finite difference of the underlying PDEs or based on finite dimensional optimization; both require discretization of the underlying space (Peyré, 2015; Carlier et al., 2017; Li et al., 2020; Carrillo et al., 2021). The computational complexity of these methods scales exponentially as the problem dimension, making them unsuitable for the cases where probability densities over high dimensional spaces are involved.

Our goal is to develop a scalable method to compute the Wasserstein gradient flow without discretizing the underlying space. One target application we are specifically interested in is optimization over the space of probability densities. Many problems such as variational inference can be viewed as special cases of such optimization. We aim to establish a method for this type of optimization that is applicable to a large class of objective functions.

Our algorithm is based on the JKO scheme (Jordan et al., 1998), which is essentially a backward Euler time discretization method for the continuous time Wasserstein gradient flow. In each step of JKO scheme, one needs to find a probability density that minimizes a weighted sum of the Wasserstein distance (square) to the probability density at the previous step and the objective function. We reparametrize this problem in each step so that the optimization variable becomes the optimal transport map from the probability density at the previous step and the one we want to optimize, recasting the problem into a stochastic optimization framework. This transport map can either be modeled by a standard feedback forward network or the gradient of an input convex neural network. The latter is justified by the fact that the optimal transport map for the optimal transport problem with quadratic cost with any marginals is the gradient of a convex function. Another crucial ingredient of our algorithm is a variational form of the objective function, which allows the evaluation of the

objective with samples and without density estimation. At the end of the algorithm, a sequence of transport maps connecting an initial distribution and the target distribution are obtained. One can then sample from the target distribution by sampling from the initial distribution (often Gaussian) and then propagating these particles through the sequence of transport maps. When the transport map is modeled by the gradient of an input convex neural network, one can evaluate the target density at every point.

Our contributions can be summarized as follows.
i). We develop a neural network based algorithm to compute Wasserstein gradient flow without spatial discretization. Our algorithm is applicable to any objective function that has a variational representation.
ii). We specialize our algorithm to three important cases where the objective functions are the Kullback-Leibler divergence, the generalized entropy, and the interaction energy .
iii). We apply our algorithm to several representative problems including sampling and aggregation-diffusion equation and obtain respectable performance.

**Related works:** Most existing methods to compute Wasserstein gradient flow are finite difference based (Peyré, 2015; Carlier et al., 2017; Li et al., 2020; Carrillo et al., 2021). These methods require spatial discretization and are thus not scalable to high dimensional settings. Salim et al. (2020) analyze the convergence for a forward-backward scheme but leave the implementation of JKO an open question. There is a line of research that uses particle-based method to estimate the Wasserstein gradient flow (Carrillo et al., 2019b; Frogner & Poggio, 2020). In these algorithms, the current density value is often estimated using kernel method whose complexity scales at least quadratically with the number of particles. More recently, three interesting neural network based methods (Mokrov et al., 2021; Alvarez-Melis et al., 2021; Yang et al., 2020) were proposed for Wasserstein gradient flow. The first one (Mokrov et al., 2021) focuses on the special case with Kullback-Leibler divergence as objective function. The second one (Alvarez-Melis et al., 2021) uses a density estimation method to evaluate the objective function by back-propagating to the initial distribution, which could become a computational burden when the number of time discretization is large. The third one (Yang et al., 2020) is based on a forward Euler time discretization of the Wasserstein gradient flow and is more sensitive to time stepsize. Over the past few years, many neural network based algorithms have been proposed to compute optimal transport map or Wasserstein barycenter (Makkuva et al., 2020; Korotin et al., 2019; Fan et al., 2020; Korotin et al., 2021). These can be viewed as special cases of Wasserstein gradient flows or optimizations over the space of probability densities.

## 2 BACKGROUND

### 2.1 OPTIMAL TRANSPORT AND WASSERSTEIN DISTANCE

Given two probability distributions $P, Q$ over the Euclidean space $\mathbb{R}^n$ with finite second moments, the optimal transport problem with quadratic cost reads

$$\min_{T: T \sharp P = Q} \int_{\mathbb{R}^n} \|x - T(x)\|_2^2 dP(x), \tag{1}$$

where the minimization is over all the feasible transport maps that transport mass from distribution $P$ to distribution $Q$. The feasibility is characterized by the pushforward operator (Bogachev, 2007) as $T \sharp P = Q$. When the initial distribution $P$ admits a density, the above optimal transport problem (1) has a unique solution and it is the gradient of a convex function, that is,

$$T^\star = \nabla \varphi$$

for some convex function $\varphi(\cdot) : \mathbb{R}^n \to \mathbb{R}$. In this paper, we assume probability measures admit densities and use the notation for the measure and the density interchangeably.

The square-root of the minimum transport cost, namely, the minimum of (1), defines a metric on the space of probability distributions known as the Wasserstein-2 distance (Villani, 2003), denoted by $W_2(P, Q)$. The Wasserstein distance has many nice geometrical properties compared with other distances such as $L_2$ distance for probability distributions, making it a popular choice in applications.

## 2.2 WASSERSTEIN GRADIENT FLOW

Given a function $\mathcal{F}(P)$ over the space of probability densities, the Wasserstein gradient flow describes the dynamics of the probability density when it follows the steepest descent direction of the function $\mathcal{F}(P)$ with respect to the Wasserstein metric $W_2$. The Wasserstein gradient flow can be explicitly represented by the PDE

$$\frac{\partial P}{\partial t} = \nabla \cdot \left( P \nabla \frac{\delta \mathcal{F}}{\delta P} \right),$$

where $\delta \mathcal{F}/\delta P$ stands for the gradient of the function $\mathcal{F}$ with respect to the standard $L_2$ metric (Villani, 2003, Ch. 8)

Many important PDEs are the Wasserstein gradient flow for minimizing certain objective functions $\mathcal{F}(P)$. For instance, when $\mathcal{F}$ is the free energy $\mathcal{F}(P) = \int_{\mathbb{R}^n} P(x) \log P(x) dx + \int_{\mathbb{R}^n} V(x) P(x) dx$, the gradient flow is the Fokker-Planck equation (Jordan et al., 1998) $\frac{\partial P}{\partial t} = \nabla \cdot (P \nabla V) + \Delta P$. When $\mathcal{F}$ is the generalized entropy $\mathcal{F}(P) = \frac{1}{m-1} \int_{\mathbb{R}^n} P^m(x) dx$ for some positive number $m > 1$, the gradient flow is the porous medium equation (Otto, 2001; Vázquez, 2007) $\frac{\partial P}{\partial t} = \Delta P^m$.

## 3 METHODS AND ALGORITHMS

We are interested in solving the optimization problem

$$\min_P \mathcal{F}(P) \tag{2}$$

over the space of probability densities $\mathcal{P}(\mathbb{R}^n)$. In particular, our objective is to develop a particle-based Wasserstein gradient flow algorithm to numerically solve (2).

The objective function $\mathcal{F}(P)$ could exhibit different form depending on the application. In this paper, we present our algorithm for the linear combination of the following three important cases:

**Case I** The functional is equal to the KL-divergence with respect to a given target distribution $Q$

$$\mathcal{F}(P) = \mathcal{D}(P||Q) := \int \log \left( \frac{P(x)}{Q(x)} \right) P(x) dx. \tag{3}$$

This is important for the problem of sampling from a target distribution.

**Case II** The objective functional is equal to the generalized entropy

$$\mathcal{F}(P) = \mathcal{G}(P) := \frac{1}{m-1} \int P^m(x) dx.$$

This case is important for modeling the porous medium.

**Case III** The objective functional is equal to the interaction energy

$$\mathcal{F}(P) = \mathcal{W}(P) := \int \int W(x-y) P(x) P(y) dx dy, \quad W : \mathbb{R}^n \to \mathbb{R}.$$

This case is important for modeling the aggregation equation.

These functionals have been widely studied in the Wasserstein gradient flow literature (Carlier et al., 2017; Santambrogio, 2017; Ambrosio et al., 2008) due to their desirable properties. It can be shown that if $\mathcal{F}(P)$ is composed by the above functionals, under proper assumptions, Wasserstein gradient flow associated with $\mathcal{F}(P)$ converges to the unique solution to (2) (Santambrogio, 2017).

In Section 3.1, 3.2, we first assume $\mathcal{F}(P)$ doesn't include interaction energy, and introduce JKO/backward scheme to solve (2). We then add $\mathcal{W}(P)$ into consideration and present a forward-backward scheme in Section 3.3 and close by showing our Algorithm in Section 3.4.

## 3.1 JKO SCHEME AND REPARAMETRIZATION

To realize the Wasserstein gradient flow, a discretization over time is needed. One such discretization is the famous JKO scheme (Jordan et al., 1998)

$$P_{k+1} = \arg\min_P \frac{1}{2a} W_2^2 (P, P_k) + \mathcal{F}(P). \tag{4}$$

This is essentially a backward Euler discretization or a proximal point method with respect to the Wasserstein metric. The solution to (4) converges to the continuous-time Wassrstein gradient flow when the step size $a \to 0$.

In our method, we reparametrize (4) as an optimization in terms of the transport maps $T : \mathbb{R}^n \to \mathbb{R}^n$ from $P_k$ to $P$, i.e., by defining $P = T\sharp P_k$. With this reparametrization, in view of the definition of Wasserstein distance (1), the JKO step (4) becomes

$$P_{k+1} = T_k\sharp P_k, \quad T_k = \arg\min_T \frac{1}{2a} \int_{\mathbb{R}^n} \|x - T(x)\|_2^2 dP_k(x) + \mathcal{F}(T\sharp P_k). \tag{5}$$

The optimal $T$ is the optimal transport map from $P_k$ to $T\sharp P_k$ and is thus the gradient of a convex function $\varphi$. Therefore, the JKO scheme can be also expressed as

$$P_{k+1} = \nabla\varphi_k\sharp P_k, \quad \varphi_k = \arg\min_{\varphi \in \text{CVX}} \frac{1}{2a} \int_{\mathbb{R}^n} \|x - \nabla\varphi(x)\|_2^2 dP_k(x) + \mathcal{F}(\nabla\varphi\sharp P_k). \tag{6}$$

where CVX stands for the space of convex functions. We use the preceding two schemes (5) and (6) in our numerical method depending on the application.

## 3.2 $\mathcal{D}(P\|Q)$ AND $\mathcal{G}(P)$ REFORMULATION WITH VARIATIONAL FORMULA

The main challenge in implementing the JKO scheme is to evaluate the functional $\mathcal{F}(P)$ in terms of samples from $P$. We achieve this goal by using a variational formulation of $\mathcal{F}$. In order to do so, we use the notion of $f$-divergence between the two distributions $P$ and $Q$:

$$D_f(P\|Q) = \mathbb{E}_Q\left[f\left(\frac{P}{Q}\right)\right] \tag{7}$$

where $f : (0, +\infty) \to \mathbb{R}$ is a convex function. The $f$-divergence admits the variational formulation

$$D_f(P\|Q) = \sup_h \mathbb{E}_P[h(X)] - \mathbb{E}_Q[f^*(h(Y))]. \tag{8}$$

where $f^*(y) = \sup_{x \in \mathbb{R}}[xy - f(x)]$ is the convex conjugate of $f$. The variational form has the special feature that it does not involve the density of $P$ and $Q$ explicitly and can be approximated in terms of samples from $P$ and $Q$. The functionals $\mathcal{D}(P\|Q)$ and $\mathcal{G}(P)$ can both be expressed as $f$-divergence.

With the help of the $f$-divergence variational formula, when $\mathcal{F}(P) = \mathcal{D}(P\|Q)$ or $\mathcal{G}(P)$, the JKO scheme (5) can be equivalently expressed as

$$P_{k+1} = T_k\sharp P_k, \quad T_k = \arg\min_T \left\{\frac{1}{2a}\mathbb{E}_{P_k}[\|X - T(X)\|^2] + \sup_h \mathcal{V}(T, h)\right\}. \tag{9}$$

where $\mathcal{V}(T, h) = \mathbb{E}_{P_k}[\mathcal{A}(T, h)] - \mathbb{E}_\Gamma[\mathcal{B}(h)]$, $\Gamma$ is a user designed distribution which is easy to sample from, and $\mathcal{A}$ and $\mathcal{B}$ are functionals whose form depends on the functional $\mathcal{F}$. The form of these two functionals for the KL divergence and the generalized entropy appears in Table 1. The details appear in Section 3.2.1 and 3.2.2.

Table 1: Variational formula for $\mathcal{D}(P\|Q)$ and $\mathcal{G}(P)$

| Energy function | $\mathcal{A}(T, h)$ | $\mathcal{B}(h)$ | $\Gamma$ |
|---|---|---|---|
| $\int P \log(P/Q) dx$ | $\log h(T) + \log \mu(T) - \log Q(T)$ | $h$ | Gaussian distribution $\mu$ |
| $\frac{1}{m-1}\int P^m dx$ | $\frac{1}{\Omega_k^{m-1}} \cdot \frac{m}{m-1}(h(T))^{m-1}$ | $\frac{1}{\Omega_k^{m-1}}h^m$ | Uniform distribution $Q$ |

### 3.2.1 KL DIVERGENCE

The KL divergence is the special instance of the $f$-divergence obtained by replacing $f$ with $f_1(x) = x \log x$ in (7)

$$D_{f_1}(P\|Q) = \mathbb{E}_Q\left[\frac{P}{Q}\log\frac{P}{Q}\right] = \mathbb{E}_P\left[\log\frac{P}{Q}\right].$$

**Proposition 1.** *The variational formulation for $\mathcal{D}(P\|Q)$ reads*

$$D_{f_1}(P\|Q) = 1 + \sup_h \mathbb{E}_P \left[ \log h(X) + \log \frac{\mu(X)}{Q(X)} \right] - \mathbb{E}_\mu \left[ h(Z) \right],$$

*where $\mu$ is a user designed distribution which is easy to sample from. The optimal function $h$ is equal to the ratio between the densities of $T\sharp P_k$ and $\mu$.*

The proof for Proposition 1 can be found in appendix A. It becomes practical when we have only access to un-normalized density of $Q$, which is the case for the sampling problem. Using this variational form in the JKO scheme (5) yields $P_{k+1} = T_k \sharp P_k$ and

$$T_k = \arg \min_T \max_h \mathbb{E}_{P_k} \left[ \frac{1}{2a} \|X - T(X)\|^2 + \log h(T(X)) + \log \frac{\mu(T(X))}{Q(T(X))} \right] - \mathbb{E}_\mu \left[ h(Z) \right] (10)$$

In practice, we choose $\mu = \mu_k$ adaptively, where $\mu_k$ is the Gaussian with the same mean and covariance as $P_k$. We noticed that this choice improves the numerical stability of the the algorithm.

### 3.2.2 POROUS MEDIUM EQUATION

The generalized entropy can be also represented as $f$-divergence. In particular, let $f_2(x) = \frac{1}{m-1}(x^m - x)$ and let $Q$ be the uniform distribution on a set which is the superset of the support of density $P(x)$ and has volume $\Omega$. Then

$$D_{f_2}(P\|Q) = \frac{\Omega^{m-1}}{m-1} \int P^m(x)dx - \frac{1}{m-1}.$$

**Proposition 2.** *The variational formulation for $\mathcal{G}(P)$ reads*

$$\mathcal{G}(P) = \frac{1}{\Omega^{m-1}} \sup_h \left( \mathbb{E}_{P_k} \left[ \frac{m}{m-1} h^{m-1}(X) \right] - \mathbb{E}_Q \left[ h^m(Z) \right] \right). \tag{11}$$

*The optimal function $h$ is equal to the ratio between the densities of $T\sharp P_k$ and $Q$.*

The proof for Proposition 2 is postponed to appendix A. Using this in the JKO scheme yields $P_{k+1} = T_k \sharp P_k$, and

$$T_k = \arg \min_T \max_h \frac{1}{2a} \mathbb{E}_{P_k} \|X - T(X)\|^2 + \frac{1}{\Omega_k^{m-1}} \left( \mathbb{E}_{P_k} \left[ \frac{m}{m-1} h^{m-1}(X) \right] - \mathbb{E}_Q \left[ h^m(Z) \right] \right),$$

where $\Omega_k$ is the volume of a set large enough to contain the support of $T\sharp P_k$ for any $T$ that is not too far away from the identity map.

---

**Algorithm 1** Primal-dual gradient flow

---

**Input:** Objective function $\mathcal{F}(P)$, initial distribution $P_0$, step size $a$, number of JKO steps $K$, number of outer loop $J_1$, number of inner loop $J_2$, batch size $M$.
**Initialization:** Parameterized $T_\theta$ and $h_\lambda$
**for** $k = 1, 2, \ldots, K$ **do**
    $P_k \leftarrow (I - a\nabla_x(W * P_k))\sharp P_k$ if $\mathcal{F}(P)$ includes $\mathcal{W}(P)$
    $T_\theta \leftarrow T_{k-1}$ if $k > 1$     {// use last iteration $T_{k-1}$ as a warm-up}
    **for** $j_1 = 1, 2, \ldots, J_1$ **do**
        Sample $Y_1, \ldots, Y_M$ from $P_k$. Sample $Z_1, \ldots, Z_M$ from $\Gamma$.
        **for** $j_2 = 1, 2, \ldots, J_2$ **do**
            Apply Adam to $\lambda$ to maximize $\frac{1}{M} \sum_{i=1}^M [\mathcal{A}(T_\theta, h_\lambda(Y_i)) - \mathcal{B}(h_\lambda(Z_i))]$
        **end for**
        Apply Adam to $\theta$ to minimize $\frac{1}{M} \sum_{i=1}^M \left[ \frac{1}{2a} \|Y_i - T_\theta(Y_i)\|^2 + \mathcal{A}(T_\theta, h_\lambda(Y_i)) \right]$
    **end for**
    $T_k \leftarrow T_\theta$
**end for**
**Output:** $\{T_k\}_{k=1}^K$

---

### 3.3 Forward Backward (FB) scheme

When $\mathcal{F}(P)$ involves the interaction energy $\mathcal{W}(P)$, we add an additional forward step to solve the gradient flow:

$$P_{k+\frac{1}{2}} := (I - a\nabla_x(W * P_k))\sharp P_k \tag{12}$$

$$P_{k+1} := T_{k+\frac{1}{2}}\sharp P_{k+\frac{1}{2}}, \tag{13}$$

where $I$ is the identity map, and $T_{k+\frac{1}{2}}$ is defined by replacing $k$ by $k + \frac{1}{2}$ in (9). In other words, the first gradient descent step (12) is a forward discretization of the gradient flow and the second JKO step (13) is a backward discretization. $\nabla_x(W * P)$ can be written as expectation $\mathbb{E}_{y \sim P}\nabla_x(W(x - y))$, thus can also be approximated by samples. Salim et al. (2020) firstly propose this method to solve Wasserstein gradient flow and provide the theoretical convergence analysis. We make this scheme practical by giving a scalable implementation of JKO.

Since $\mathcal{W}(P)$ can be equivalently written as expectation $\mathbb{E}_{x,y \sim P}[W(x - y)]$, there exists another non-forward-backward (non-FB) method , i.e., removing the first step and integrating $\mathcal{W}(P)$ into a single JKO step: $P_{k+1} = T_k \sharp P_k$ and

$$T_k = \arg\min_T \left\{ \frac{1}{2a}\mathbb{E}_{P_k}\|X - T(X)\|^2 + \mathbb{E}_{X,Y \sim P_k}[W(T(X) - T(Y))] + \sup_h \mathcal{V}(T, h) \right\}.$$

In practice, we observe the FB scheme is more stable however converge slower than non-FB scheme. The detailed discussion appears in the Appendix B.2, B.3.

**Remark 1.** *In principle, one can single out $\log(Q)$ term from (10) and perform a similar forward step $P_{k+\frac{1}{2}} = (I - a(\nabla_x Q)/Q)\sharp P_k$ (Salim et al., 2020), but we don't observe improved performance of doing this in sampling task.*

### 3.4 Primal-dual algorithm and parametrization of $T$ and $h$

The two optimization variables $T$ and $h$ in our minimax formulation (9) can be both parameterized by neural networks, denoted by $T_\theta$ and $h_\lambda$. With this neural network parametrization, we can then solve the problem by iteratively updating $T_\theta$ and $h_\lambda$. This prime-dual method to solve (2) is depicted in Algorithm 1.

In this work, we implemented two different architectures for the map $T$. One way is to use a neural network to represent $T$ directly, and another way is to parametrize $T$ as the gradient of a Input convex neural network (ICNN) (Amos et al., 2017) $\varphi$. The latter has been widely used in optimal transport (Makkuva et al., 2020; Fan et al., 2020; Korotin et al., 2021). In our experiments, we find that the first parameterization gives better result in the sampling application. As we discuss in Section 3.5, when density evaluation is needed, we adopt the ICNN parameterization since we need to compute $T^{-1}$. Note that ICNN could be modified to be strictly convex and if the function $\varphi$ is strictly convex, the gradient $\nabla\varphi$ is invertible.

### 3.5 Evaluation of the density

In this section, we assume the solving process doesn't use forward-backward scheme, i.e. all the probability measures $P_k$ are obtained by performing JKO one by one. Otherwise, the map $I - a\nabla_x(W * P_k) = I - \mathbb{E}_{y \sim P_k}\nabla_x(W(x - y))$ includes an expectation term and becomes intractable to push-backward particles to compute density.

If $T$ is invertible, these exists a standard approach to evaluate the density of $P_k$ (Alvarez-Melis et al., 2021; Mokrov et al., 2021) through the change of variables formula. More specifically. we assume $T$ is parameterized by the gradient of an ICNN $\varphi$ that is assumed to be strictly convex. To evaluate the density $P_k(x_k)$ at point $x_k$, we back propagate through the sequence of maps $T_k = \nabla\varphi_k, \ldots, T_1 = \nabla\varphi_1$ to get

$$x_i = T_{i+1}^{-1} \circ T_{i+2}^{-1} \circ \cdots \circ T_k^{-1}(x_k).$$

The inverse map $T_j^{-1} = (\nabla\varphi_j)^{-1} = \nabla\varphi_j^*$ can be obtained by solving the convex optimization

$$x_{j-1} = \arg\max_{x \in \mathbb{R}^n}\langle x, x_j\rangle - \varphi_j(x). \tag{14}$$

Then, by the change of variables formula, we obtain

$$\log[P_k(x_k)] = \log[P_0(x_0)] - \sum_{i=1}^{k} \log \left| \nabla^2 \varphi_i(x_{i-1}) \right|, \tag{15}$$

where $\nabla^2 \varphi_i(x_{i-1})$ is the Hessian of $\varphi_i$ and $\left| \nabla^2 \varphi_i(x_{i-1}) \right|$ is its determinant. By iteratively solving (14) and plugging the resulting $x_j$ into (15), we can recover the density $P_k(x_k)$ at any point.

### 3.6 COMPUTATIONAL COMPLEXITY

Per each update $k$ in Algorithm 1, the forward step (12) requires at most $O(N^2)$ where $N$ is the total number of particles to push-forward. The backward step (13) requires $O(J_1 k M H)$ where $J_1$ is the number of iterations per each JKO step, $M$ is the batch size, and $H$ is the size of the network. $k$ shows up in the bound because sampling $P_k$ requires us to pushforward $x_0 \sim P_0$ through $k-1$ maps.

However, both Mokrov et al. (2021) and Alvarez-Melis et al. (2021) require $O(J_1 k (nMH + n^3))$ which has the cubic dependence on dimension $n$ because they need to query the $\log \det \nabla^2 \varphi$ in each iteration. We refer to Mokrov et al. (2021, Section 5) for the complexity details of calculating the Hessian term. Thus our method has the advantage of independence on the dimension.

We provide training time details in Section 4.2 and Appendix B.4. Other than training and sampling time, the complexity for evaluating the density are the same as the above two methods due to the standard density evaluation process (see Section 3.5).

## 4 NUMERICAL EXAMPLES

### 4.1 SAMPLING

We first consider the sampling problem to sample from a target distribution $Q$. Note that $Q$ doesn't have to be normalized. To this end, we consider the Wasserstein gradient flow with objective function $\mathcal{F}(P) = \int P \log(P/Q) dx$, that is, the KL divergence between distributions $P$ and $Q$. When this

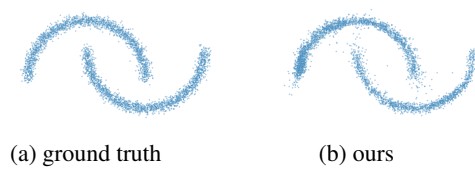

(a) ground truth        (b) ours

Figure 1: The left figure shows samples from the target 16-GMM distribution and the right figure shows samples obtained by our method. Each plot contains 4000 points.

objective is minimized, $P \propto Q$. In our experiments, we consider two types of target distribution: the two moons distribution and the Gaussian mixture model (GMM) with spherical Gaussian components. In this set of experiments, the step size is set to be $a = 0.3$ and the initial measure is a spherical Gaussian $\mathcal{N}(0, 2.25 I_d)$.

**Two moons:** The two moons distribution is a popular target distribution for sampling task. It is a 2D mixture model composed of 16 Gaussian components; each moon shape consists of 8 Gaussian components. The results are displayed in Figure 1, from which we see that our method is able to generate samples that match the target distribution.

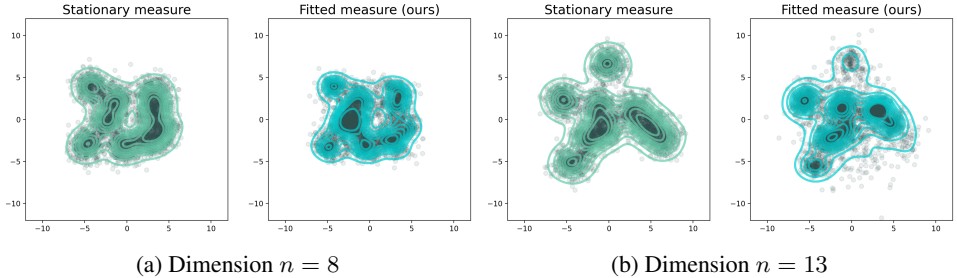

(a) Dimension $n = 8$                    (b) Dimension $n = 13$

Figure 2: Comparison between the target GMM and fitted measure of generated samples by our method. Samples are projected onto 2D plane by performing PCA. We refer the reader to Mokrov et al. (2021) for the performance of another algorithm in similar setup.

**GMM with spherical Gaussians:** We also test our algorithm in sampling from GMM in higher dimensional space. The target GMM has 9 Gaussian components with equal weights and the same covariances. The results with dimension $n = 8$ and $n = 13$ are depicted in Figure 2. In Figure 2, we not only display the samples as grey dots in the plot, but also the kernel density estimation of generated samples as level sets. As can be seen from the results, both the samples and densities obtained with our algorithm match the target distribution well.

## 4.2 Ornstein-Uhlenbeck Process

We study the performance of our method in modeling the Ornstein-Uhlenbeck Process as dimension grows. The gradient flow is affiliated with the free energy (3), where $Q = e^{(x-b)^{\mathsf{T}} A(x-b)/2}$ with a positive definite matrix $A \in \mathbb{R}^n \times \mathbb{R}^n$ and $b \in \mathbb{R}^n$. Given an initial Gaussian distribution $\mathcal{N}(0, I_n)$, the gradient flow at each time $t$ is a Gaussian distribution $P_t$ with mean vector $\mu_t = (I_n - e^{-At})b$ and covariance $\Sigma_t = A^{-1}(I_n - e^{-2At}) + e^{-2At}$ (Vatiwutipong & Phewchean, 2019). We calculate $P_k$ with JKO step size $a = 0.05$ and compare with the Fokker-Planck (FP) JKO (Mokrov et al., 2021). We quantify the error as the SymKL divergence between estimated distribution and the ground truth in Figure 3, where

$$\mathrm{SymKL}(P1, P2) := D(P_1 \| P_2) + D(P_2 \| P_1).$$

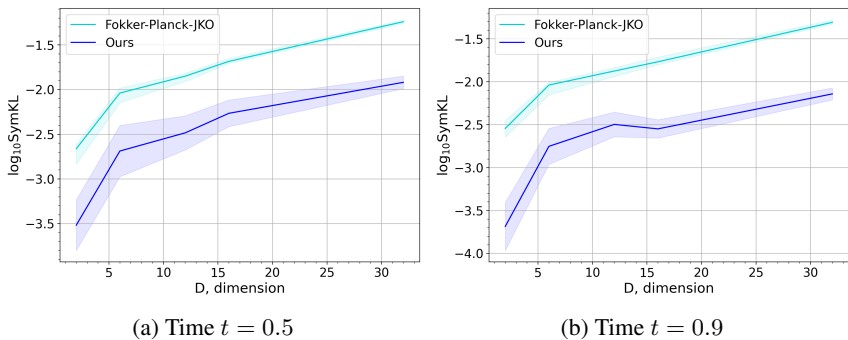

(a) Time $t = 0.5$      (b) Time $t = 0.9$

Figure 3: We repeat the experiments for 15 times in dimensions $d = 2, 6, 12, 16, 32$.

We also compare the training time per every two JKO steps with FP JKO. The computation time for FP JKO is around $20s$ when $d = 2$ and increases to $100s$ when $d = 32$. Our method's training time remains at $20s \pm 2s$ for all the dimensions $d = 2 \sim 32$. This is due to we fix the neural network size for both methods and our method's computation complexity doesn't depend on the dimension.

## 4.3 Porous media equation

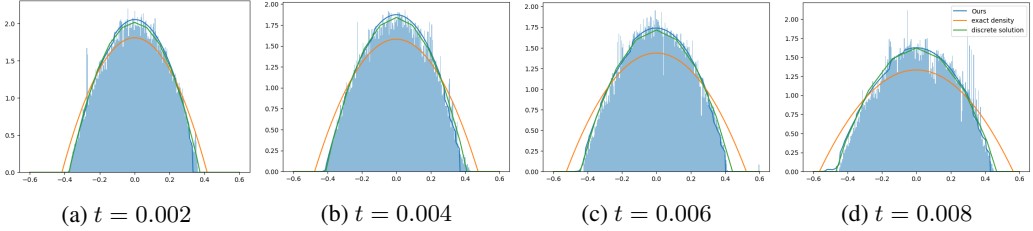

(a) $t = 0.002$      (b) $t = 0.004$      (c) $t = 0.006$      (d) $t = 0.008$

Figure 4: Comparison among exact density, finite difference method solution given by `CVXOPT`, and the density given by our method. To better visualize the distributed particles from each distribution, we also plot the histograms of our method as the blue shadow.

We next consider the porous media equation with only diffusion: $\partial_t P = \Delta P^m$. This is the Wasserstein gradient flow associated with the energy function $\mathcal{F}(P) = \frac{1}{m-1} \int P^m(x)dx$. A representative

closed-form solution of the porous media equation is the Barenblatt profile (GI, 1952; Vázquez, 2007)

$$P_{gt}(t,x) = (t+t_0)^{-\alpha}\left(C - \beta\left(x - x_0\right)^2\left(t+t_0\right)^{-2\alpha/n}\right)_+^{\frac{1}{m-1}},$$

$$\text{where}\quad \alpha = \frac{n}{n(m-1)+2}, \quad \beta = \frac{(m-1)\alpha}{2mn},$$

and $t_0 > 0$ is the starting time and $C > 0$ is a free parameter.

In principle, our algorithm should match the analytical solution $P_{gt}$ when the step size $a$ is sufficiently small. When $a$ is not that small, time discretization is inevitable. To account for the time-discretization error of JKO scheme, we consider the porous media equation in 1D space and use the solution via finite difference method as a reference. The details appear in appendix C.3.

In the experiments, we set the stepsize for the JKO scheme to be $a = 0.001$ and the initial time to be $t_0 = 0.002$. Other parameters are chosen as $C = (3/16)^{1/3}, m = 2, n = 1, d = 300$. We parametrize the transport map $T$ as the gradient of an ICNN and thus we can evaluate the density following Section 3.5. In Figure 4, we observe that the gap between the density computed using our algorithm and the ground truth density $P_{gt}$ is dominated by the time-discretization error of the JKO scheme. Our result matches the discrete time solution nicely.

## 4.4 Aggregation–Diffusion Equation

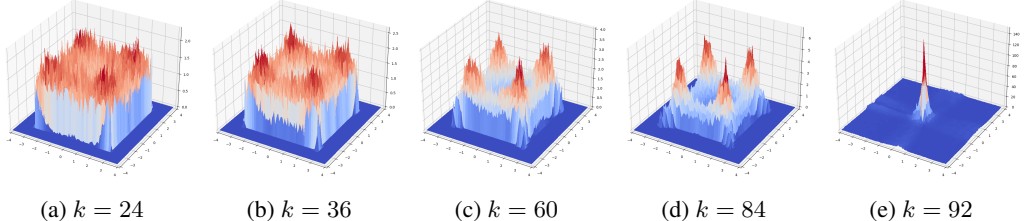

(a) $k = 24$     (b) $k = 36$     (c) $k = 60$     (d) $k = 84$     (e) $k = 92$

Figure 5: Histogram for simulated measures $P_k$ by FB scheme at different $k$.

We finally simulate the evolution of solutions to the following aggregation-diffusion equation:

$$\partial_t P = \nabla \cdot (P \nabla W * P) + 0.1\Delta P^m, \quad W(x) = -e^{-\|x\|^2}/\pi.$$

This corresponds to the energy function $\mathcal{W}(P) + 0.1\mathcal{G}(P)$. We use the same parameters in Carrillo et al. (2021, Section 4.3.3). The initial distribution is a uniform distribution supported on $[-3,3] \times [-3,3]$ and the JKO step size $a = 0.5$. In Figure 5, we utilize FB scheme to simulate the gradient flow for this equation with $m = 3$ on $\mathbb{R}^2$ space. With this choice $W(x)$, $\nabla_x(W * P_k)$ is equal to $\mathbb{E}_{y \sim P_k}\left[2e^{-\|x-y\|^2}/\pi\right]$ in the gradient descent step (12). And we estimate $\nabla_x(W * P_k)$ with $10^4$ samples from $P_k$.

Throughout the process, the aggregation term $\nabla \cdot (P \nabla W * P)$ and the diffusion $0.1\Delta P^m$ adversarially exert their effects and cause the probability measure split to four pulses and converge to a single pulse in the end (Carrillo et al., 2019a).

## 5 Conclusion

In this paper we presented a novel neural network based algorithm to compute the Wasserstein gradient flow. Our algorithm follows the JKO time discretization scheme. We reparametrize the problem so that the optimization variable becomes the transport map $T$ between the consecutive steps. By utilizing a variational formula of the objective function, we further reformulate the problem in every step as a min-max problem over map $T$ and dual function $h$ respectively. This formulation doesn't require density estimation using samples and can be optimized using stochastic optimization. It also shows advantages with dimension-independent computation complexity. Our method can also be extended to minimize other objective functions that can be written as $f$-divergence. Our limitation is the accuracy is not satisfying in sampling tasks with high dimension complex density.

## 6 REPRODUCIBILITY

The basic setup is distributed in Section 4. We refer to Section 3.4 and Appendix C for all the rest training details. The code of our method is also attached in the supplementary material.

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

## A  DETAILS ABOUT VARIATIONAL FORMULA IN SECTION 3.2

### A.1  KL DIVERGENCE

The KL divergence is the special instance of the $f$-divergence obtained by replacing $f$ with $f_1(x) = x \log x$ in (7)

$$D_{f_1}(P\|Q) = \mathbb{E}_Q \left[ \frac{P}{Q} \log \frac{P}{Q} \right] = \mathbb{E}_P \left[ \log \frac{P}{Q} \right],$$

which, according to (8), admits the variational formulation

$$D_{f_1}(P\|Q) = 1 + \sup_h \mathbb{E}_P [h(X)] - \mathbb{E}_Q \left[ e^{h(Y)} \right] \tag{16}$$

where the convex conjugate $f_1^*(y) = e^{y-1}$ is used.

The variational formulation can be approximated in terms of samples from $P$ and $Q$. For the case where we have only access to un-normalized density of $Q$, which is the case for the sampling problem, we use the following change of variable: $h \to \log(h) + \log(\mu) - \log(Q)$ where $\mu$ is a user designed distribution which is easy to sample from. Under such a change of variable, the variational formulation reads

$$D_{f_1}(P\|Q) = 1 + \sup_h \mathbb{E}_P \left[ \log h(X) + \log \frac{\mu(X)}{Q(X)} \right] - \mathbb{E}_\mu [h(Z)].$$

Note that the optimal function $h$ is equal to the ratio between the densities of $T\sharp P_k$ and $\mu$.

**Remark 2.** *The Donsker-Varadhan formula*

$$\mathcal{D}(P\|Q) = \sup_h \mathbb{E}_P [h(X)] - \log \mathbb{E}_Q \left[ e^{h(Z)} \right]$$

*is another variational representation of KL divergence and it's a stronger than* (16) *because it's a upper bound of* (16) *for any fixed h. However, we cannot get an unbiased estimation of the objective using samples.*

### A.2  POROUS MEDIUM EQUATION

The generalized entropy can be also represented as $f$-divergence. In particular, let $f_2(x) = \frac{1}{m-1}(x^m - x)$ and let $Q$ be the uniform distribution on a set which is the superset of the support of density $P(x)$ and has volume $\Omega$. Then

$$D_{f_2}(P\|Q) = \frac{\Omega^{m-1}}{m-1} \int P^m(x) dx - \frac{1}{m-1}.$$

As a result, the generalized entropy can be expressed in terms of $f$-divergence according to

$$\mathcal{G}(P) = \frac{1}{m-1} \int P^m(x) dx = \frac{1}{\Omega^{m-1}} D_{f_2}(P\|Q) + \frac{1}{\Omega^{m-1}(m-1)}.$$

Upon using the variational representation of the $f$-divergence with

$$f_2^*(y) = \left( \frac{(m-1)y+1}{m} \right)^{\frac{m}{m-1}},$$

the generalized entropy admits the following variational formulation

$$\mathcal{G}(P) = \sup_h \frac{1}{\Omega^{m-1}} \left( \mathbb{E}_P[h(X)] - \mathbb{E}_Q \left[ \left( \frac{(m-1)h(Y)+1}{m} \right)^{\frac{m}{m-1}} \right] \right) + \frac{1}{\Omega^{m-1}(m-1)}.$$

In practice, we find it numerically useful to let $h = \frac{1}{m-1} \left[ m \left( \hat{h} \right)^{m-1} - 1 \right]$ so that

$$\mathcal{G}(P) = \frac{1}{\Omega^{m-1}} \sup_{\hat{h}} \left( \mathbb{E}_{P_k} \left[ \frac{m}{m-1} \hat{h}^{m-1}(X) \right] - \mathbb{E}_Q \left[ \hat{h}^m(Z) \right] \right).$$

With such a change of variable, the optimal function $\hat{h} = T\sharp P_k/Q$.

# B ADDITIONAL EXPERIMENT RESULTS AND DISCUSSIONS

## B.1 SAMPLING USING ICNN PARAMETERIZATION

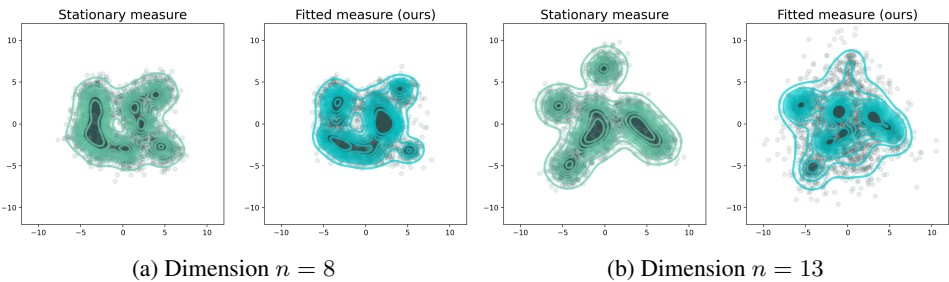

(a) Dimension $n = 8$      (b) Dimension $n = 13$

Figure 6: Sampling Gaussian mixture models by parameterizing the map by $\nabla\varphi$.

In Figure 6, we present the sampling results with $\nabla\varphi$ parameterized map where $\varphi$ is a ICNN neural network. The experiment setting is the same as Section 4.1 and we can observe a MLP network map gives better fitted measures.

## B.2 AGGREGATION EQUATION

Alvarez-Melis et al. (2021) proposes using the neural network based JKO, i.e. the backward method, to solve (17). They parameterize $T$ as the gradient of the ICNN. In this section, we use two cases to compare the forward method and backward when $\mathcal{F}(P) = \mathcal{W}(P)$. This could help explain the FB and non-FB scheme performance difference later in Section B.3.

We study the gradient flow associated with the aggregation equation

$$\partial_t P = \nabla \cdot (P \nabla W * P), \quad W : \mathbb{R}^n \to \mathbb{R}. \tag{17}$$

The forward method is

$$P_{k+1} := (I - a\nabla_x(W * P_k))\sharp P_k.$$

The backward method or JKO is

$$P_{k+1} := T_k \sharp P_k, \quad T_k = \arg\min_T \left\{ \frac{1}{2a}\mathbb{E}_{P_k}[\|X - T(X)\|^2] + \mathbb{E}_{X,Y \sim P_k}[W(T(X) - T(Y))] \right\}.$$

**Example 1** We follow the setting in Carrillo et al. (2021, Section 4.3.1 ). The interaction kernel is $W(x) = \frac{\|x\|^4}{4} - \frac{\|x\|^2}{2}$, and the initial measure $P_0$ is a Gaussian $\mathcal{N}(0, 0.25I)$. In this case, $\nabla_x(W * P_k)$ becomes $\mathbb{E}_{y \sim P_k}\left[(\|x - y\|^2 - 1)(x - y)\right]$. We use step size $a = 0.05$ for both methods and show the results in Figure 7.

**Example 2** We follow the setting in Carrillo et al. (2021, Section 4.2.3 ). The interaction kernel is $W(x) = \frac{\|x\|^2}{2} - \ln\|x\|$, and the initial measure $P_0$ is $\mathcal{N}(0, 1)$. The unique steady state for this case is

$$P_\infty(x) = \frac{1}{\pi}\sqrt{(2 - x^2)_+}.$$

The reader can refer to Alvarez-Melis et al. (2021, Section 5.3) for the backward method performance. As for the forward method, $\nabla_x(W * P_k)$ becomes $\mathbb{E}_{y \sim P_k}\left[x - y - \frac{1}{x-y}\right]$. Because the kernel $W$ enforces repulsion near the origin and $P_0$ is concentrated around origin, $\nabla_x(W * P)$ will easily blow up. So the forward method is not suitable for this kind of interaction kernel.

Through the above two examples, if $\nabla_x(W * P)$ is smooth, we can notice the backward method converges faster, but is not stable when solving (17). This shed light on the FB and non-FB scheme performance in Section 4.4, B.3. However, if $\nabla_x(W * P)$ has bad modality such as Example 2, the forward method loses the competitivity.

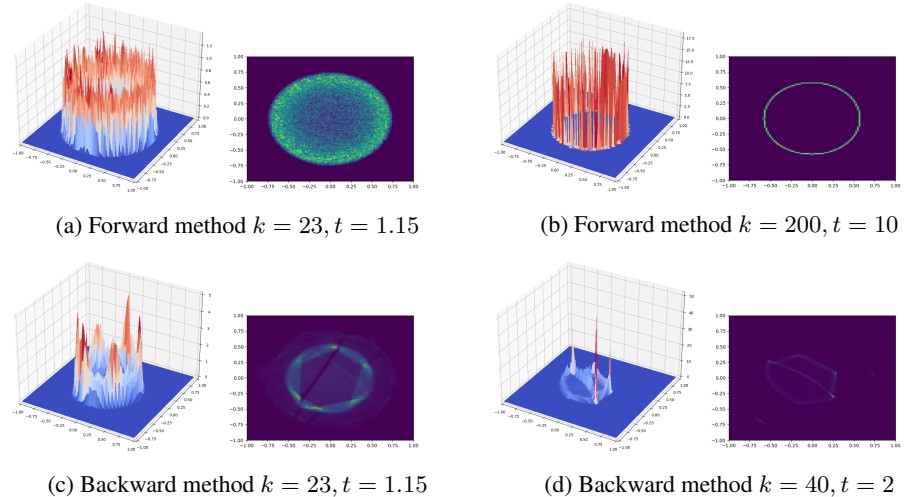

(a) Forward method $k = 23, t = 1.15$      (b) Forward method $k = 200, t = 10$

(c) Backward method $k = 23, t = 1.15$      (d) Backward method $k = 40, t = 2$

Figure 7: The steady state is supported on a ring of radius 0.5. Backward converges faster to the steady rate but is unstable. As $k$ goes large, it cannot keep the regular ring shape and will collapse after $k > 50$.

### B.3 AGGREGATION-DIFFUSION EQUATION WITH NON-FB SCHEME

In Figure 8, we show the non-FB solutions to Aggregation-diffusion equation in Section 4.4. FB scheme should be independent with the implementation of JKO, but in the following context, we assume FB and non-FB are both neural network based methods discussed in Section 3. Non-FB scheme reads

$$P_{k+1} = T_k \sharp P_k$$
$$T_k = \arg\min_T \left\{ \frac{1}{2a} \mathbb{E}_{P_k}[\|X - T(X)\|^2] + \mathbb{E}_{X,Y \sim P_k}[W(T(X) - T(Y))] + \mathcal{G}(T, h) \right\},$$

where $\mathcal{G}(T, h)$ is represented by the variational formula (11). We use the same step size $a = 0.5$ and other PDE parameters as in Section 4.4.

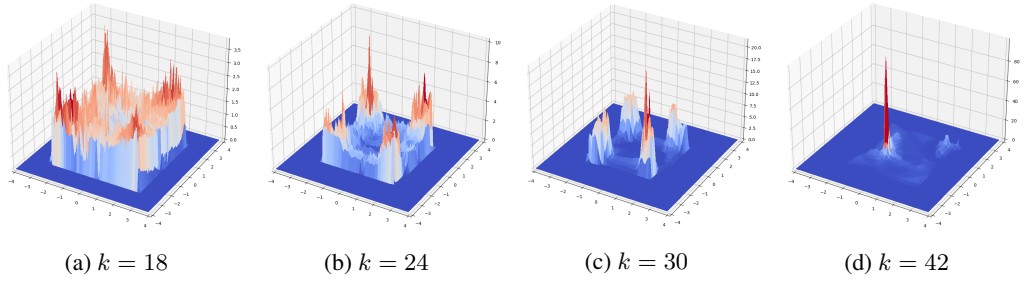

(a) $k = 18$      (b) $k = 24$      (c) $k = 30$      (d) $k = 42$

Figure 8: Histograms for simulated measures $P_k$ by non-FB scheme at different $k$.

Comparing the FB scheme results in Figure 5 and the non-FB scheme results in Figure 8, we observe non-FB converges $1.5\times$ slower than the finite difference method (Carrillo et al., 2021), and FB converges $3\times$ slower than the finite difference method. This may because splitting one JKO step to the forward-backward two steps removes the aggregation term effect in the JKO, and the diffusion term is too weak to make a difference in the loss. Note at the first several $k$, both $P_k$ and $Q$ are nearly the same uniform distributions, so $h$ is nearly a constant and $T(x)$ exerts little effect in the variational formula of $\mathcal{G}(P)$. Another possible reason is a single forward step for aggregation term converges slower than integrating aggregation in the backward step, as we discuss in Section B.2 and Figure 7.

However, FB generates more regular measures. We can tell the four pulses given by FB are more symmetric. We speculate this is because gradient descent step in FB utilizes the geometric structure of $W(x)$ directly, but integrating $\mathcal{W}(P)$ in neural network based JKO losses the geometric meaning of $W(x)$.

### B.4 COMPUTATIONAL TIME

Our experiments are conducted on GeForce RTX 3090. The forward step (12) takes about 14 seconds to pushforward one million points.

Assume each JKO step involves 500 iterations and the number of inner iteration $J_1 = 3$, then each JKO step (13) takes 100 seconds if the energy function contains the generalized energy $\mathcal{G}(P)$ and 25 seconds if the energy function contains the KL divergence $\mathcal{D}(P\|Q)$.

## C  IMPLEMENTATION DETAILS

Our code is written in Pytorch-Lightning (Falcon & Cho, 2020). For some parts of plotting in Section 4.1 and 4.2, we adopt the code given by Mokrov et al. (2021).

Without further specification, we use the following parameters:
1) The number of iterations of the outer loop $J_1$ is 600.
2) The number of iterations of the inner loop $J_2$ is 3.
3) The batch size is fixed to be $M = 100$.
4) The learning rate is fixed to be 0.001.
5) All the activation functions are set to be PReLu.
6) $h$ has 4 layers and 16 neurons in each layer.
7) $T$ has 5 layers and 16 neurons in each layer.

The transport map $T$ can be parametrized in different ways. We use a residual MLP network for it in Section 4.1, 4.2, B.2 and the gradient of a strongly convex ICNN in Section 4.3, 4.4, B.1, B.3. The dual test function $h$ is always a MLP network with a dropout layer before each layer.

### C.1  SAMPLING (SECTION 4.1 AND B.1)

**Two moons**   We run $K = 10$ JKO steps with $J_2 = 6$ inner iterations. $h$ has 6 layers. $T$ has 5 layers.

**GMM**   8D example trains for $K = 50$ JKO steps. $h$ has 6 layers and 64 neurons in each layer. $T$ has 3 layers and 128 neurons in each layer.

13D example trains for $K = 20$ JKO steps. $h$ has 8 layers and 64 neurons in each layer. $T$ has 9 layers and 64 neurons in each layer.

### C.2  ORNSTEIN-UHLENBECK PROCESS (SECTION 4.2)

For Fokker-Planck JKO, we use the implementation provided by the authors and the default parameters given in Mokrov et al. (2021, Section A.2). We also estimate the SymKL using Monte Carlo according to the author's instructions.

For our method, we use a linear residual feed-forward NN to work as $T$, i.e. without activation function. $h$ and $T$ both have 3 layers and 64 hidden neurons per layer for all dimensions. We also train them for $J_1 = 500$ iterations per each JKO with learning rate 0.005. The batch size is $M = 1000$.

However, we estimate SymKL for our algorithm in a different way. Since our map $T$ is a linear transformation, our estimated $\widetilde{P}_t$ is guaranteed to be a Gaussian distribution. We firstly draw $5 \cdot 10^5$ samples from $\widetilde{P}_t$ and calculate the empirical mean $\widetilde{\mu}_t$ and covariance $\widetilde{\Sigma}_t$. Then we estimate $\mathcal{D}(\widetilde{P}_t\|P_t)$ using the following closed form KL divergence between two Gaussians

$$\mathcal{D}(\widetilde{P}_t\|P_t) = \frac{1}{2}\left[\log\frac{|\Sigma_t|}{|\widetilde{\Sigma}_t|} - d + (\widetilde{\mu}_t - \mu_t)^{\mathrm{T}}\,\widetilde{\Sigma}_t^{-1}(\widetilde{\mu}_t - \mu_t) + \mathrm{tr}(\Sigma_t^{-1}\widetilde{\Sigma}_t)\right].$$

$\mathcal{D}(P_t\|\widetilde{P}_t)$ is estimated similarly.

### C.3 POROUS MEDIA EQUATION (SECTION 4.3)

In the experiment, $h$ and $T$ both have 10 neurons in each layer.

To account for the time-discretization error of JKO scheme, we consider the porous media equation in 1D space and use the solution via finite difference method as a reference. More specifically, in the 1D space $\mathbb{R}$, we discretize the density over a fixed grid with grid size $d$ and grid resolution $\delta x$. With this discretization, the probability densities become (scaled) probability vectors and the problem (4) can be converted into a convex optimization

$$\min_{\pi:\delta x \pi^{\mathrm{T}} \mathbf{1}=\widehat{P}_k} \frac{(\delta x)^2}{2a} \langle \pi, M \rangle + \frac{\delta x}{m-1}(\mathbf{1}^{\mathrm{T}}\pi\delta x)^m \mathbf{1} \tag{18}$$

where $M$ is the discretized unit transport cost, $\widehat{P}_k \in \mathbb{R}^d$ is the probability vector at the previous step, $\mathbf{1} \in \mathbb{R}^d$ is the all-ones vector and the optimization variable $\pi \in \mathbb{R}^d \times \mathbb{R}^d$ is the joint distribution between $\widehat{P}_k$ and $\widehat{P}_{k+1}$. This is a standard convex optimization and can be solved with generic solvers. When an optimal $\pi$ is obtained, $\widehat{P}_{k+1}$ can be computed as $\widehat{P}_{k+1} = \pi\mathbf{1}$. We adopt the library CVXOPT[1] to solve the convex programming problem (18). In so doing, we arrive at a reference solution $\widehat{P}_0, \widehat{P}_1, \ldots, \widehat{P}_k, \ldots$ for our algorithm.

### C.4 AGGREGATION-DIFFUSION EQUATION (SECTION 4.4 AND B.3)

Each JKO step contains $J_1 = 200$ iterations. The batch size is $M = 1000$.

---

[1]http://cvxr.com/cvx/

