# OpenReview forum: "Variational Wasserstein gradient flow"
_ICLR.cc/2022/Conference — ICLR 2022 Submitted_

### Official Review · Reviewer_CWde · 2021-10-20

**Correctness:** 2
**Technical Novelty And Significance:** 2
**Empirical Novelty And Significance:** 2
**Recommendation:** 3
**Confidence:** 5

**Main Review:**

**Benefits**
1) The variational approach might be a solution to issues of direct computation such as cubically growing complexity as a function of the dimension;

**Drawbacks**

2) The experiments of the paper are weak and do not sufficiently support the main claims;
3) The relation to the prior work is not fully disclosed;
4) The scope of the paper (WGFs) is narrow.

*Detailed comments are below.*

**Relation to prior work.** A large part of the algorithm proposed by the authors matches the previously known art. More precisely, sections 3.0 and 3.1 exploit the reformulation of JKO via functions that have already been proposed in [1] (not cited in the paper!) and extensively used in [2,3]. I think this is not very clear from the text and might add extra (inexistent) value to the current work.

If I correctly understand, the actual difference w.r.t. the prior work, e.g., [2], is the way that the f-divergence is optimized. The authors use a variational approximation in contrast to direct computation. Here I have two remarks.

First, the key claim of the paper is that direct computation scales cubically and is not feasible in high dimensions. I agree, but what about fast approximations? In [3] the authors explicitly state they use a fast estimator based on the Hutchinson method. In [2] the authors state that to speed up the computation fast approximations can be used. It is unfair to ignore this and compare in the experimental section only with [2] using and only by using the direct computation.

Second, the variational approximation of the f-divergence is not novel, see, for example, f-GANs [5]. I wonder if such approximations have already been used in Bayesian machine learning (BML)? I encourage the authors to include a detailed discussion of this. Why is this important? I am not an expert in BML, but it seems to me that this particular variational approximation can be applied to a dozen of tasks involving KL divergence. The current paper demonstrates that it outperforms direct computation. If this is indeed true, why hasn’t this approach been used for other BML tasks? If it has already been, the authors should include relevant references to support the current experimental findings.

Overall, the paper should more carefully acknowledge the prior work and make it transparent what is new and what is already well known. Paper [4] also seems relevant.

**Experiments.** The experimental section is very weak both in terms of quality and quantity. In terms of quantity, the paper looks poor compared to predecessors, e.g., [2], a paper to which they compare their method. In particular, the results provided in Figure 2 visually suffer from notable artifacts, which is a contrast to that reported by [2] in a similar setup (Figure 2 of [2]). Besides, the dimensions considered in the current paper are lower, which is suspicious.

Importantly, the quantitative results provided in Figure 3 raise questions. How is it possible that on such a toy example (evolving Gaussian distributions -- linear pushforward maps grad psi), the variational methods so drastically outperform the direct computation (up to 10 times)? This is quite unbelievable, in particular, in small dimensions. Did the authors use the same network architectures and other shared hyperparameters for comparison? A discussion here is needed.

**Scope.** If I correctly understand, the authors do not provide any high-dimensional applications of WGFs. Due to this, I currently tend to think the scope of the paper is narrow and the usefulness of the proposed approach for the community of ICLR is questionable. Adding an application would definitely benefit the paper.

**Correctness of the method.** Overall, the method is correct. However, in one of the experiments, the authors approximate the pushforward of the JKO step directly as the neural network, not as the gradient of the input-convex network. In this case, the push forward distribution might not have density damaging the entire JKO scheme.

**Clarity.** What for the FB scheme is introduced (section 3.3)? This is not clear from the text.

**References**

[1] Benamou, J. D., Carlier, G., Mérigot, Q., & Oudet, E. (2016). Discretization of functionals involving the Monge–Ampère operator. Numerische mathematik, 134(3), 611-636.

[2] Mokrov, P., Korotin, A., Li, L., Genevay, A., Solomon, J., & Burnaev, E. (2021). Large-Scale Wasserstein Gradient Flows. arXiv preprint arXiv:2106.00736.

[3] Alvarez-Melis, D., Schiff, Y., & Mroueh, Y. (2021). Optimizing Functionals on the Space of Probabilities with Input Convex Neural Networks. arXiv preprint arXiv:2106.00774.

[4] Bunne, C., Meng-Papaxanthos, L., Krause, A., & Cuturi, M. (2021). JKOnet: Proximal Optimal Transport Modeling of Population Dynamics. arXiv preprint arXiv:2106.06345.

[5] Nowozin, S., Cseke, B., & Tomioka, R. (2016, December). f-gan: Training generative neural samplers using variational divergence minimization. In Proceedings of the 30th International Conference on Neural Information Processing Systems (pp. 271-279).

**Summary Of The Paper:**

The paper proposes a method to compute Wasserstein Gradient Flows (WGFs) via neural networks and the JKO scheme. In contrast to prior works, to compute WGFs of functionals involving f-divergences, the authors use variational approximations rather than direct computations. It is claimed to work faster and perform better.

**Summary Of The Review:**

My overall impression of the paper is that it is unfinished. While the idea of variational approximation is reasonable, I suppose this paper requires a major revision with a dozen text improvements and experiments. Therefore, I vote to reject this paper in its current form.

---

> ### Author Response · Authors · 2021-11-22
> **reply**
>
> Thank you for your suggestion about BML. There exist some works in BML that leverage the variational formula of f-divergence, e.g., Wan, Neng, Dapeng Li, and Naira Hovakimyan. "f-Divergence Variational Inference." Advances in Neural Information Processing Systems 33 (2020). We would add this as a discussion.
>
> FB scheme is introduced when the target functional includes the case III interaction energy functional. The forward step makes our solution more stable.

---

### Official Review · Reviewer_S3mH · 2021-11-01

**Correctness:** 4
**Technical Novelty And Significance:** 2
**Empirical Novelty And Significance:** 2
**Recommendation:** 3
**Confidence:** 4

**Main Review:**

This paper identifies a crucial challenge in the existing works of emulating JKO steps, namely the expensive computation of the densities of the form $P_k(x)$ for each $k$. The solution the paper suggests is reasonable, but it comes at the cost of adding another inner maximization which makes the optimization a lot more difficult (e.g. unstable due to high variance). Overall I think the amount of contribution is very limited. The variational formulation of the objective functionals considered is all well known, and putting it together with the JKO step is fairly straightforward. The experiments are not convincing enough to demonstrate the practical advantages of the proposed method compared to the alternatives.

Detailed comments:
- In Eq (9), a distribution $\Gamma$ is introduced. What is the point of introducing this measure? Is it essentially an importance sampling of $Q$ for which we only know the density? Why is it enough to just choose Gaussians, which could be very different from $Q$?
- Many sections are very similar to [Mokrov 2021]. For example, there is nothing new in Sec 3.5, and the experiment's setup in 4.1 and 4.2 are exactly the same. Yet the more challenging experiments from [Mokrov 2021] are not reproduced here, such as posterior inference and non-linear filtering.
- In Sec 3.6, it is common to use the Hutchinson trace estimator to approximate the gradient of $\log \det$ (in addition to a linear solve), which could speed up the competing methods. It might be good to include a comparison to that.
- The results in Figure 2 are visibly worse than those of [Mokrov 2021]. Moreover, here only up to dimension 13 is included, whereas [Mokrov 2021] contains dimension 32.
- The results in Figure 3 of the proposed method are better than those of [Mokrov 2021]. I'm wondering why this is the case since in [Mokrov 2021] the KL divergence is calculated exactly, whereas in the proposed method additional bias could be introduced due to the failure of maximizing $h$.


**Summary Of The Paper:**

This paper proposes a variational formulation of each JKO step for optimizing functionals on measures.
Different from existing recent works on emulating JKO steps by training pushforward neural networks (either directly or as gradients of convex functions), the variational formulation involves another inner maximization of a function, without needing density access that typically requires cubic time complexity due to computing the log determinants of the pushforwards. Experiments are done to demonstrate the practicality of the algorithm.


**Summary Of The Review:**

The paper studies an important challenge in JKO steps encountered by recent works, but the contributions are incremental without demonstrating convincing practical advantages.

---

### Official Review · Reviewer_72fj · 2021-11-02

**Correctness:** 4
**Technical Novelty And Significance:** 1
**Empirical Novelty And Significance:** 3
**Recommendation:** 3
**Confidence:** 4

**Main Review:**

Strengths:

The paper is well written and high level ideas on the theory are explained. The approach is clear and reasonable. The simulations show promising results, they do not seem to have been cherry picked. They show what is claimed by the authors and there is no surprise given the simplicity of the approach.

Weaknesses:

The theoretical consistence of the proposed methods is ignored. Are the proposed scheme consistent time discretizations of the WGF? Are the assumptions satisfied? Is the alternative maximization/minimization strategy (Algo 1) consistent? Do all measures considered admit density wrt Lebesgue? The paper mainly provides intuition for Algo 1 without solid theoretical foundations.

Moreover, the technical contribution is rather limited (see the summary of the paper above). One could argue that it is an "easy paper" in the sense that only the simulations seem new. The novelty is mainly to have parametrized the functions in the objective of the JKO by neural networks.

**Summary Of The Paper:**

This paper studies the implementation of some Wasserstein Gradient Flows (WGF) in discrete time but without discretizing the space. The methods proposed are based on the JKO operator to discretize WGF in time. The implementation of the JKO can be challenging. The strategy of the authors is to first reparametrize the JKO as a minimization over a space of functions (instead of measures) via pushforward. Then, when the objective function is a f-divergence, the objective inside the JKO admit a variational representation and can be expressed as a sup. Conclusion: each JKO is written as a min max over a space of functions. To solve it, they parametrize the functions by neural networks and alternatively maximize and minimize the problem using Adam. An important feature is that the objective in the min max can be approximated with samples of the current distribution (its density doesn't appear, only integrals wrt to the current distribution).

**Summary Of The Review:**

The approach is reasonable, the simulation promising but I do not see a significant technical contribution (They essentially reparametrized a min max problem over a function space using neural nets and running Adam to alternatively maximize and minimize).

---

> ### Author Response · Authors · 2021-11-22
> **reply**
>
> We agree that adding some theoretical study will make the paper more solid. We will add some theoretical results such as consistency in the revision.

---

> > ### Comment · Reviewer_72fj · 2021-11-22
> > **Thanks**
> >
> > Thanks for your reply.
> >
> > I believe that the points raised in the comments need to be improved/fixed before publication. Hence, the paper is not ready yet and needs to be revised.

---

### Official Review · Reviewer_pvCT · 2021-11-02

**Correctness:** 3
**Technical Novelty And Significance:** 2
**Empirical Novelty And Significance:** 1
**Recommendation:** 5
**Confidence:** 4

**Main Review:**

Strengths:
* An ingenious use of f-divergence duality (aka variational formulation) to re-write density-depending functional flow objectives, such as the generalized entropy, as (optimization of) expectations, allowing for computation via finite-sample approximation
* The resulting method seems to be empirically valid in low- and mid-dimensional settings

Weaknesses:
* Novelty, the main (and perhaps, only) novelty of this approach compared to (Mokrov et al., 2021; Alvarez-Melis et al., 2021; Bunne et al., 2021) is the use of the variational formulation of f-divergences, something that is itself well known. Note that this is only relevant for functional objectives that depend on the density of the measure itself, e.g., those that cannot be expressed as an expectation over the measure. All other objectives, including the interaction energy considered here, can be tackled with the exact same approach of the three papers above.
* The novel contributions should be more clearly separated from non-novel ones. E.g., the reformulation of WGF as optimization over convex functions might seem like a contribution of this paper to an uninitiated reader, although it has been explored extensively in other work, most prominently in Benamou et al (2014), which is puzzling not cited here, and more recently in the 3 JKO + ICNN papers cited above.
* Some statements are not clear, not correct or too vague. These should all be clarified or corrected. For example:
    * "that is not too far away from the identity map" in Pg 5.
    * In section 2.2, the $\delta F / \delta P$ is not formally a gradient, but the first-variation of a functional
    * In Section 3.6, it is stated that prior works requires explicit computation of log determinants of Hessians at cubic complexity, but this is not the case. At least one of those methods uses cuadrtic-cost (matrix-vector-product) stochastic estimators of the Hess logdet.
* The motivation for using the FB scheme for the interaction energy is not clear. Some of the intuition/motivation discussed in sections B.2 and B.3 should probably be moved to the main text.
* Limited experimental validation, especially with regards to high-dimensional settings, which is arguably the main promise of this work. In addition, all experiments are synthetic. A more compelling evaluation framework should include at least one high-dimensional, realistic dataset.
* The evaluation is mostly qualitative. The paper purposefuly chooses PDEs with known solutions, but then provides mostly qualitative/visual results. A more compelling evaluation framwork would include quantitative comparison against these known solutions in sections 4.3 and 4.4.
* While the computational complexity summary provided in the paper is useful, there are some many hidden constants in those bounds that they are hardly useful. These should be complemented with a thorough empirical runtime analysis comparing against exact and inexact methods, such as those cited here as related work.

Minor comments:
* While the motivation is different, there are deep and unexplored connections between the dualizatin of the f-divergence objectives used here and the dual of the kantorovich OT problem that has been recently explored extensively for learning Monge maps between distributions. Many of these rely on convex conjugacy to reformulate a sup objective as a sup-inf one (see Korotin et al. 2021b for an excellent survey on these). It would have been great (though not obligatory!) to see a discussion on these connections
* I would suggest moving Table 1 after Proposition 1, so that all objects in the definition of $\mathcal{A}(T,h)$ have been already introduced. I spent some time trying to figure out what $\mu(T)$ was before finding it further down.

Missing related work:

* Benamou et al, "Discretization of functionals involving the Monge-Ampère operator". 2014.
* Huang et al, "Convex Potential Flows: Universar Probability Distributions with Optimal Transport and Convex Optimization", ICLR 2021.
* Bunne et al., "JKOnet: Proximal Optimal Transport Modeling of Population Dynamics", 2021.

**Summary Of The Paper:**

This paper proposes a method to solve Wasserstein gradient flows based on the JKO scheme using variational formulations of functional objectives, such as the KL divergence or the generalized entropy (non-linear diffusion). Relying on known reformulations of the JKO scheme as optimization over convex functions, the paper departs from recent related methods in expressing certain objectives as f-divergences, and in turn using the dual formulation of these divergences to circumvent the need to do explicit density computation in these. The resulting method involves parametrizing two types of operators as neural networks (one of them as an input-convex neural network), and solving a mini-max objective. The paper presents experiments on simple PDEs (mostly in 1D or 2D) with known solutions.

**Summary Of The Review:**

The paper provides an interesting variation on recent JKO-based methods for computational Wasserstein gradient flows, but its limited novelty and empirical evaluation diminish its contribution, and make it a borderline paper in my view. That being said, if the issues I raise in my review are properly addressed, I would be willing to increase my score.

---

### Author Response · Authors · 2021-11-22
**Replies to reviewers.**

We would like to thank the reviewers for the comments and suggestions. We would modify the paper according to the comments.

**We first clarify our contributions here:**

- The JKO reformulation in Section 3.1 is already proposed in several existing works and is not part of our contribution (except for the idea of using a neural network without ICNN structure constraint to parameterize the map). We will clarify this and add relevant references in the revision.

- The forward-backward scheme is proposed in Salim et al. 2020. Section 3.5 is also discussed in both Alvarez-Melis et al. 2021 and Mokrov et al. 2021. They are not parts of the contribution in this paper. We add them to the paper to make the structure complete. We will clarify this in the revision.

- Our main contribution is that we use f-divergence to obtain the variational formula of interested functionals so that when functional objectives cannot be expressed as an expectation over the measure (typically internal energy), we can still approximate the functional with samples.

**Computational complexity of log det |Hessian|.**

There exists a fast approximation of log det |Hessian| and its gradient using Stochastic Lanczos Quadrature and Hutchinson trace estimator. The cubic dependence on dimension can be improved to quadratic dependence. However, this is accompanied by an additional cost, which is the number of iterations to run the conjugate gradient (CG) method. CG is guaranteed to converge exactly in $n$ steps (n is the dimension). If we want to obtain $\nabla$ log det |Hessian| precisely, the cost is still $O(n^3)$, which is the same as calculating  $\nabla$ log det |Hessian| directly. If we use an error $\epsilon$ stopping condition in CG, the complexity could be improved to $\sqrt{\kappa \log (2 / \epsilon) n^2}$, where $\kappa$ is the conditional number of Hessian but this would sacrifice the accuracy.

**Results in Figure 3:**

We use nearly all the same hyper-parameters including learning rate, hidden layer width, and the number of iterations per JKO step. We believe that there are several reasons we have better performance, 1) the proposed distribution $\mu$ is Gaussian, which is consistent with $P_t$ for any $t$. This is beneficial for the inner maximization to find a precise $h$. 2) the map $T$ is linear, so it can promise our generated samples are always from a Gaussian distribution. Note that if one approximates the map as the gradient of ICNN, it's not guaranteed that the mapped distribution is still Gaussian.


**density of measures**

We agree that the push-forwarded distribution might not have density. There exists some work in normalizing flow literature to guarantee the invertible deep neural networks and thus one can calculate the density of push-forwarded distribution in a tractable way.
But it's worth mentioning that those neural networks with strong Lipschitz constraints would narrow down the expression ability of NNs.  Recently some paper also proposes to replace the gradient of ICNN with a neural network to increase the expression ability to better approximate the map between two distributions, e.g.

Rout, Litu, Alexander Korotin, and Evgeny Burnaev. "Generative Modeling with Optimal Transport Maps." *arXiv preprint arXiv:2110.02999* (2021).

In the examples we present, $T(x)$ gives more regular solutions compared to ICNN based parameterization. If $T$ is not invertible, $T(x)$ still possibly has a density with a large chance; the problem is that the density would be very difficult to compute.
So if we don't add a strong Lipschitz constraint for $T$, our method would become a particle-based method, i.e. we cannot query density directly.

---

### Decision · Program_Chairs · 2022-01-20

**Decision:**

Reject

**Comment:**

The paper proposes  a min/max reformulation for JKO gradient flows appealing the variational formulation of f-divergences. This would alleviate the need of an explicit density. All reviewers pointed out the limited novelty in the work and the limited experimentation.

We encourage authors to add a theoretical analysis to their work and further strengthening of the experimental section with high dimensional experiments, and to resubmit the work on an upcoming venue.